# Generation of interspecies limited chimeric nephrons using a conditional nephron progenitor cell replacement system

S. Yamanaka[1], S. Tajiri[1,2], T. Fujimoto[1,2], K. Matsumoto[1], S. Fukunaga[1,3], B.S. Kim[2,4], H.J. Okano[2] & T. Yokoo [1]

Animal fetuses and embryos may have applications in the generation of human organs. Progenitor cells may be an appropriate cell source for regenerative organs because of their safety and availability. However, regenerative organs derived from exogenous lineage progenitors in developing animal fetuses have not yet been obtained. Here, we established a combination system through which donor cells could be precisely injected into the nephrogenic zone and native nephron progenitor cells (NPCs) could be eliminated in a time- and tissue-specific manner. We successfully achieved removal of Six2+ NPCs within the nephrogenic niche and complete replacement of transplanted NPCs with donor cells. These NPCs developed into mature glomeruli and renal tubules, and blood flow was observed following transplantation in vivo. Furthermore, this artificial nephron could be obtained using NPCs from different species. Thus, this technique enables in vivo differentiation from progenitor cells into nephrons, providing insights into nephrogenesis and organ regeneration.

[1] Division of Nephrology and Hypertension, Department of Internal Medicine, Jikei University School of Medicine, Tokyo 1058461, Japan. [2] Division of Regenerative Medicine, Jikei University School of Medicine, Tokyo 1058461, Japan. [3] Department of Internal Medicine IV, Shimane University, Izumo, Shimane 6938501, Japan. [4] Department of Urology, Kyungpook National University School of Medicine, Daegu 41944, Korea. Correspondence and requests for materials should be addressed to T.Y. (email: tyokoo@jikei.ac.jp)

F etuses complete the complex process of nephrogenesis (kidney development) within a set time while still inside the mother's uterus[1]. Thus, renal regeneration may become feasible if the developmental program could be completely recapitulated[2]. However, the development of organs during the fetal period is subject to complex spatiotemporal regulation, making regeneration of the kidney in a dish exceedingly difficult. For this reason, we have developed a strategy for applying multipotent stem cells at the niche of organogenesis[2–8]. This strategy involves transplantation of human cells into the area of nephrogenesis in a fetus of a different animal species, thereby generating human cell-derived kidneys[2].

Gardner and Jhonson reported the generation of a rat-mouse chimera by injection of inner cell mass into blastocysts[9], demonstrating that certain differentiation signals could be shared between species. Many researchers have attempted to explore interspecies chimeras or chimeric organs using embryos and fetuses of different animals[10]. Using such technology, attempts to regenerate solid organs, such as pancreases and kidneys, in xeno-animals have recently been made using blastocyst complementation, in which embryonic stem (ES) cells or induced pluripotent (iPS) cells are injected into blastocysts lacking key molecules to generate the organ of interest[11, 12]. However, due to the pluripotency of the injected cells, their progeny may be disseminated throughout the chimera, resulting in serious ethical concerns with regard to contribution to host gametes or neural tissues. To overcome these problems, researchers have attempted to control chimerism using the *Mixl1* gene to regulate the endodermal lineage or Sox17+ endoderm progenitors injected into blastocysts expressing the anti-apoptotic gene *BCL2*[13, 14].

Additionally, injection of lineage progenitors into early-stage embryos is not sufficient for organ regeneration after implantation[10, 14, 15]. Instead, transplantation of donor cells into developmental stage-matched host embryos may be critical for the efficient engraftment of cells in generating chimeras[15]. Moreover, whole organ regeneration has not yet been performed with injection or transplantation of lineage progenitors as donor cells into the later stage fetus (postimplantation) or lineage niche[10, 15]. In this context, it is necessary to develop a system in which the organ of interest is exclusively rebuilt in the chimera using organ-specific progenitor cells.

Since the differentiation potency of progenitor cells is limited by certain lineages, they are extremely unlikely to affect the central nervous system or reproductive system across germ layers. Additionally, renal progenitor cells have recently been induced from iPS/ES cells[16–19], and advancements in technology have made large-scale culture of these cells a possibility[20, 21]. Therefore, progenitor cells may be able to serve as a promising source of regenerative organs in the future. When progenitor cells are used as donor cells, the host fetus should be in the later stages of development for stage-matching[10]. For example, nephron progenitor cells (NPCs) exist from embryonic day 10.5 (E10.5) to the early neonatal stage in mice[1, 22]. Therefore, we hypothesized that it would be optimal to transplant progenitor cells into a fetus during this specific period and assumed that this method of exploiting limited chimeras may prove useful for organ regeneration.

We previously reported that injection of human mesenchymal stem cells expressing glial-derived neurotrophic factor into the nephrogenic niche of the developing fetus may result in the formation of chimeric kidneys in the host compartment[2, 3]. However, injection of cells into the nephrogenic niche at the initial budding from the Wolffian duct before nephrogenesis is technically difficult and cannot be applied for larger animals. Therefore, it is necessary to establish a system that is technically

sound and can be applied for larger animals to efficiently inject donor cells into the nephrogenic niche.

Accordingly, in the present study, we established a technical method through which donor cells could be precisely injected at the nephrogenic zone of the mouse metanephros (MN), where injected cells are continuously developed at later stages of kidney development. The nephrogenic zone was defined as the layer under renal capsule in which there was a plural number of nephrogenic niches[23, 24]. Thus, we attempted to transplant cells NPCs, which were from developmental stage-matched host embryos, into the nephrogenic zone to form whole nephrons. Although several reports have described sporadically attempts to transplant cells into the MN[25, 26], these attempts have not been sufficiently successful, and even when nephrons have been regenerated, they have been reported to contain a mixture of host and donor cells, the latter of which should be fully replaced by native host progenitor cells to generate whole kidneys. We further established a system in which native NPCs were eliminated in both a time- and tissue-specific manner to allow external NPCs to develop to form whole nephrons and evaluated the ability of these nephrons to develop into mature glomeruli and renal tubules by integrating with the vasculature of the host animal following transplantation in vivo. We also injected rat NPCs into mouse MNs in the elimination model, generating chimeric nephrons. These results provided insights into donor NPC-derived kidney regeneration in the organ niche of another species.

## Results

**Transplantation of NPCs under the renal capsule of the MN.** To evaluate whether isolated NPCs could be integrated into the CM region, which is the origin of nephron development, we attempted transplantation into the nephrogenic zone of the MN at the same embryonic stage. First, we isolated NPCs from E13.5 embryos of mice expressing green fluorescent protein (GFP) under the control of the ubiquitously expressed CMV-β-actin promoter (CAG-GFP mice). These GFP-expressing NPCs (GFP-NPCs) were then transplanted into the E13.5 MN of B6 mice. After 5–7 days of organ culture, the samples were subjected to histological analysis via immunostaining. We transplanted the NPCs by simply applying them around the MN (outside MN group; Fig. 1a, left column). However, NPCs did not integrate into the nephrogenic zone and CM (Fig. 1b, upper and 1c, left).

The renal capsule contains a cell layer of tight junctions[27]. Therefore, we assumed that the capsule layer may prevent integration of NPCs into the CM. We attempted to transplant NPCs directly into the nephrogenic zone by inserting them under the renal capsule of the MN (Fig. 1a, right and 1b, lower). Notably, GFP-NPCs inserted into the nephrogenic zone in this group integrated into the CM region around the ureteric tips (Fig. 1c, right and 1d, right). In contrast, the outside group exhibited no GFP-positive cells in the proximity of the ureteric tips (Fig. 1c, left and 1d, left). These integrated GFP-positive cells expressed Six2 and thus were thought to constitute Six2-positive NPCs (Six2+NPCs; Fig. 1d, right and 1e). Furthermore, transplanted and integrated GFP-expressing cells were also observed on some C-shaped nephrons in the inside MN group (Fig. 1e). These cells were shown to possess the potency to differentiate into nephrons by receiving subepithelial signals (Fig. 1e). However, these early renal vesicles had a mosaic structure (Fig. 1e, yellow arrowhead and Supplementary Movie 1). Although a previous study demonstrated that NPCs transplanted into the MN can differentiate into mosaic nephrons[25], no attempts were made to transplant cells under the renal capsule. The original method facilitated the transplantation of a larger number of cells into the nephrogenic zone (Fig. 1f, right schematic and Supplementary Movie 2). In the inside wild-type

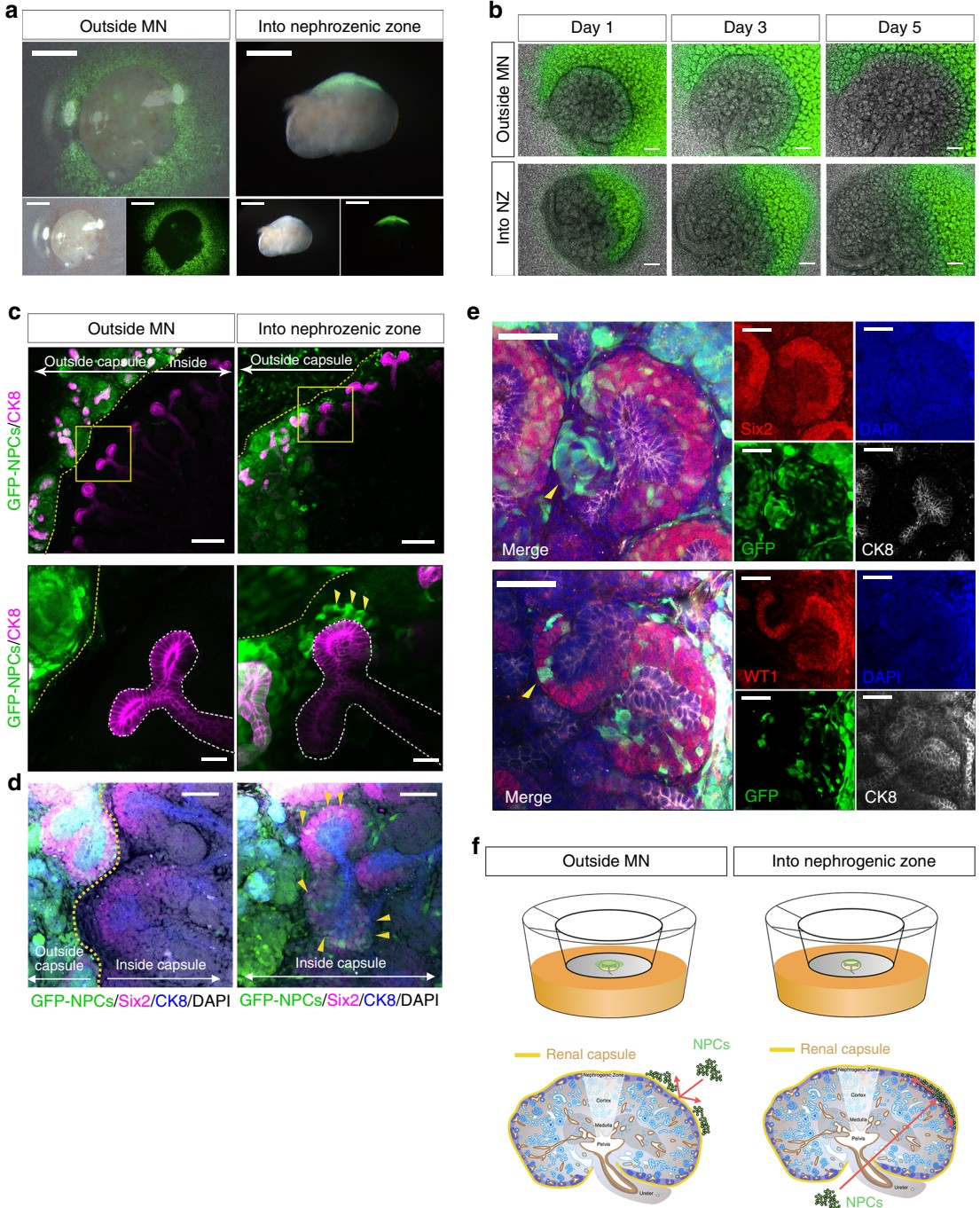

**Fig. 1** Integration of nephron progenitor cells (NPCs) into the cap mesenchyme (CM). **a** Nephron progenitor cells (NPCs) that expressed green fluorescent protein (GFP) were observed transplanted outside the metanephros (MN) or into the nephrogenic zone (NZ) under the renal capsule of E13.5 wild-type MNs (scale bar, 500 μm). **b** MN-injected NPCs were cultured at the air−fluid interface on a polycarbonate filter. The NPCs were observed under a fluorescence microscope (scale bar, 200 μm). **c** The MN was costained with cytokeratin-8 (CK8) to label the ureteric bud and GFP-NPCs of donor cells (arrows). After 5 days of culture, a few cells were integrated into the CM at the inside renal capsule of the injected NPCs (scale bar, left: 100 μm, right: 20 μm). **d** Left column: transplantation of GFP-NPCs outside the MN; right column: transplantation of GFP-NPCs into the NZ area. Integrated GFP-positive cells expressed Six2 (yellow arrow; scale bar, 50 μm). **e** Integrated GFP-NPCs were observed on some C-shaped nephrons with mosaic formation in the NZ group (scale bar, 50 μm). **f** Schematic representation of NPC transplantation into the nephrogenic zone (right) or the outside MN (left). Schematic of metanephros adapted from GUDMAP[50]

MN group, the integration rate of cells was about 30% for each ureteric tip in the CM region (Fig. 2a, Table 1). Importantly, although the integration of NPCs into the CM was good, this method did not achieve complete replacement of NPCs or regeneration of pure nephrons. Instead, this method yielded a cell composition that was a mixture of both transplanted and host-derived cells in the CM region, with the differentiated nephrons forming mosaics.

**Evaluation of the cell elimination system.** The wild-type MN occupied the CM area through host NPCs; therefore, complete

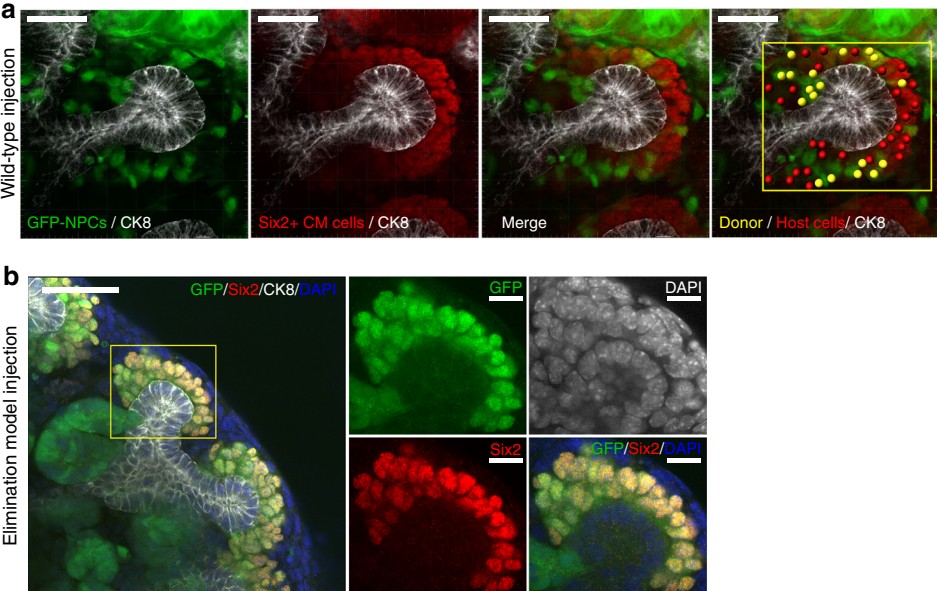

**Fig. 2** Representative images of cap mesenchyme (CM) which injected GFP nephron progenitor cells. **a** Nephron progenitor cells (NPCs)-expressing GFP were transplanted into the nephrogenic zone in the wild-type metanephros (MN). Whole-mount immunofluorescence of the cultured MN was used to count the numbers of injected NPCs remaining in the CM. Right column showed that the red spot was host NPCs (Six2 positive) and the yellow spot was donor NPCs (Six2 and GFP positive). The yellow square was an analysis area. **a** scale bar, 40 μm. **b** NPCs expressing GFP were transplanted into the nephrogenic zone in the elimination model of Six2-Cre-inducible diphtheria toxin receptor (iDTR) MN. The yellow square was CM area. Right column was high magnification views. **a** scale bar, left: 40 μm. right: 10 μm. Each MN were subjected to organ culture for 5 days

replacement of CM cells by donor cells was limited in the wild-type MN. Accordingly, we attempted to eliminate host NPCs in the CM. To design a system that could specifically eliminate all NPCs present in the CM, we hybridized Six2-GFPCre mice[22] with Cre-inducible diphtheria toxin (DT) receptor (iDTR) transgenic mice[28]. The resulting mice (Six2-GFPCre+; iDTR+ mice) are referred to as Six2-iDTR mice (Fig. 3a). Six2-GFPCre mice were heterozygotes, and iDTR+ mice were homozygotes. The Six2-iDTR embryos were obtained at expected Mendelian ratios (half ratios).

The MN isolated from each Six2-iDTR mouse was subjected to organ culture (Transwell). DT was dispensed into organ culture chambers at varying concentrations from 0.001 to 0.1 ng/μL and was administered for 5 days after the medium was replaced (Fig. 4). As a result, GFP expression was absent by day 3 at all concentrations (Fig. 4), and higher DT concentrations resulted in earlier elimination of GFP expression. When DT was administered 100 μL at 0.1 ng/μL (10 ng/well), GFP expression was eliminated by 36 h after administration (Fig. 3b). After Six2-DTR MNs were cultured from day 0 (first day of culture) to day 5 in the presence of 0.1 ng/μL DT, the CM exhibited almost no signs of Six2- or GFP-expressing cells on day 5 (Fig. 3c, lower). Compared with that in the untreated control group, the DT-treated group displayed poor branching of the ureteric bud[29], which did elongate but had an extremely deformed shape with many distortions (Fig. 3c, lower right). Continuous DT administration to the Six2-iDTR MN resulted in almost complete elimination of Six2-positive NPCs by day 5, and the expression of Six2 was abolished as early as approximately 36–48 h after DT administration (Fig. 3a, low and Fig. 4).

**Renal regeneration by the cell elimination system**. We hypothesized that we may be able to regenerate kidneys by transplanting new DT-uneffective NPCs into the nephrogenic zone simultaneously with DT-mediated elimination of host Six2-NPCs.

Accordingly, we first isolated the MN from E13.5 Six2-iDTR mice and then transplanted GFP-NPCs or DsRed-NPCs collected from E13.5 CAG-GFP mice or mice expressing DsRed under the same promoter (CAG-DsRed mice) without DT receptor (DTR) expression into the nephrogenic zone under the renal capsule (Fig. 5a, top and 5d, top). Each MN was subjected to organ culture and was administered DT at 0.1 ng/μL starting from day 0. Analysis performed on days 5–7 revealed that despite continuous administration of DT, GFP-positive or DsRed-positive transplanted cells expressing Six2 existed on the CM (Fig. 2b, Fig. 5a, top and right column and 5b and 5d, top). Post-transplanted NPCs were fully replaced in the CM of the elimination model (Figs. 2b and 5d, top right column). Sufficient replacement required elimination of host NPCs. Daily DT administration caused almost complete elimination of Six2-positive cells (Fig. 5a, second from bottom and 5d, second from top). When no DT was administered, only the Six2-positive cells in the Six2-iDTR MN expressed GFP, whereas epithelialized nephrons including glomeruli and renal tubules did not (Fig. 5a, bottom and 5d, bottom). However, when GFP-NPCs or DsRed-NPCs were transplanted into the Six2-iDTR MN along with DT administration, we observed GFP- or DsRed-expressing Wilms tumor 1 (WT1)-positive glomeruli (Fig. 5c, right, Fig. 5e, and Supplementary Movies 3–5), E-cadherin-positive distal convoluted tubules (Fig. 5e, bottom column), and *Lotus tetragonolobus* lectin (LTL)-positive proximal convoluted tubules (Fig. 5c, left). These regenerative nephron structures were shown to be connected to the collecting duct derived from the host ureteric bud (Fig. 5c, yellow arrowhead and line, Fig. 5e second from bottom, bottom column, and Supplementary Movies 5 and 6). To determine whether cell transplantation inhibited the activity of DT and impeded cell elimination from the CM, we transplanted fibroblasts extracted from DsRed mice into the Six2-iDTR MN. In this case, no Six2-positive cells occurred in the CM of the host (Fig. 6).

By eliminating existing NPCs simultaneously with cell transplantation into nephrogenic zone and replacing them with transplanted cells, we succeeded in regenerating nephrons that were completely derived from transplanted NPCs. This method was very simple and exhibited a high success rate. However, the

collecting duct and ureter were not differentiated by the transplanted cells. NPCs from the metanephric mesenchyme lineage differentiate into nephrons; however, the collecting duct and ureter are derived from the ureteric epithelium lineage[18]. This is the first report of renal regeneration focusing on Six2-positive NPCs in which donor NPCs were replaced.

**In vivo renal regeneration by the cell elimination system**. For examination of integration and connection of the vasculature in the neonephron and examination of the further development of the neonephron, we evaluated a system in which renal regeneration was mediated by the replacement of NPCs in vivo.

First, we injected GFP- or DsRed-NPCs plus DT below the renal capsule of an isolated Six2-iDTR MN (Fig. 7a, b and Supplementary Movie 2) and then transplanted the Six2-iDTR MN containing DT-uneffective exogenous NPCs into the proximity of the aorta of B6 mice (Fig. 7c, left and 7f, schematic). The transplanted MN was not separated from the ureter and bladder, but was instead isolated as a whole with these

| Table 1 Measurement of the replacement rate in the cap mesenchyme (CM) | | |
| --- | --- | --- |
| GFP-NPCs (Donor cells) | Six2 + CM cells (total cells) | Percentage of GFP cells in the Cap mensechyme |
| 11.4 ± 1.4 | 32.5 ± 2.5 | 36.9 ± 4.4 (%) |

mean±SEM (n = 12)
The fraction of Six2-positive green cells (yellow spot) integrated into the CM of the total Six2-positive cells (red and yellow spot) is presented as a percentage. Nephron progenitor cells-expressing GFP transplanted in the wild-type metanephros (MN). Representative image is Fig. 2a
Nephron progenitor cells injected into wild-type metanephros
GFP-NPC green fluorescent protein-expressing nephron progenitor cells

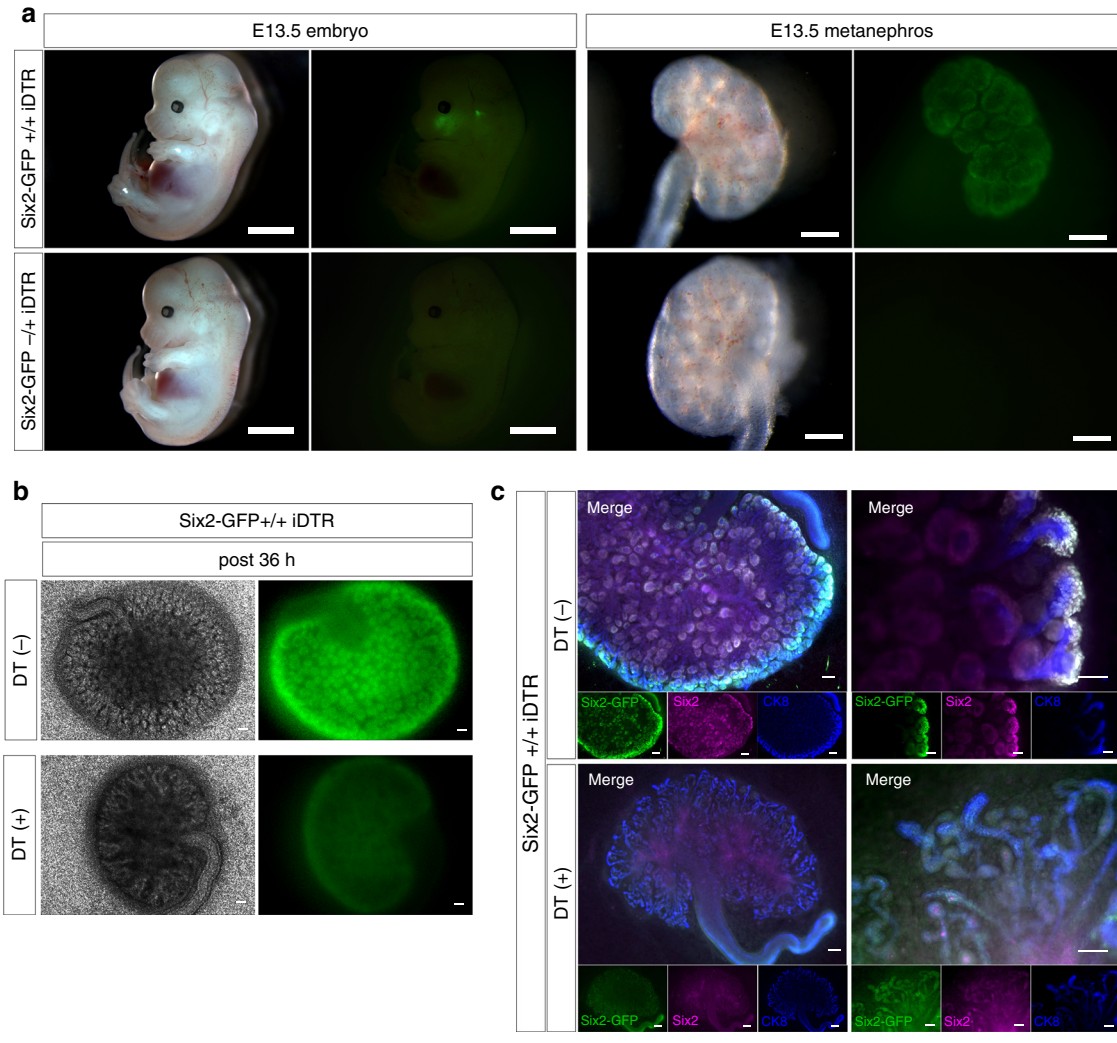

**Fig. 3** Six2-Cre-inducible diphtheria toxin receptor (iDTR) model for ablation of Six2+ cells in the cap mesenchyme (CM). **a** Generation of bigenic offspring from heterozygous Six2-GFPCre+ mice and homozygous iDTR+ mice. Inheritance of transgenes occurred at Mendelian ratios. Animals testing positive for both transgenes (Six2-GFPCre+/+ iDTR) were considered bigenic (scale bar, embryo: 1 mm, metanephros: 200 μm). **b** Thirty-six hours after the first DT administration, the progenitor elimination model displayed numerous depleted cells in the nephrogenic zone, unlike vehicle (PBS) injection (scale bar, left: 500 μm, right: 500 μm). **c** Comparison of Six2-iDTR MNs between DT- and vehicle-mediated cell elimination. DT-mediated cell elimination gave rise to apoptosis in Six2-positive nephron progenitor cells in the CM (Six2: magenta, GFP: green, lower column) but not to collecting ducts because of their ureteric bud lineage (CK-8: blue, lower column). Administration of PBS resulted in no elimination of nephron progenitor cells in the CM (upper column; scale bar, 50 μm)

**Fig. 4** Adjusting the concentration of diphtheria toxin (DT). **a-d** We evaluated cell elimination by administering a dose of diphtheria toxin (DT). Vehicle (phosphate-buffered saline (PBS))-treated Six2-iDTR mice served as controls. In the metanephros (MN) of Six2-iDTR mice, apoptotic death of Six2-GFP cells in the CM resulted from DT-mediated cell elimination at all densities (0.5 ng/well (medium, 400 μL) to 50 ng/well; scale bar, 500 μm)

components connected and subsequently transplanted as an MNB (Fig. 7a, right and 7b, right).

On day 14, the MNB was recovered and prepared as frozen sections (Fig. 7c, right and second from right), which were then subjected to immunostaining (Fig. 7d, left). As a result, GFP-NPCs-derived glomeruli were found to have formed within the transplanted Six2-iDTR MN. The glomeruli expressed GFP and had a podocin-positive loop wall (Fig. 7d, second from left, Supplementary Movie 7). CD31-positive vascular endothelial cells were found within these regenerated glomeruli (Fig. 7d, left, second from right). Furthermore, hematoxylin−eosin staining revealed the presence of erythrocytes within the glomeruli, implying their connection to host blood vessels and inflow of blood (Fig. 7, right). Electron microscopic observation of these glomeruli detected glomerular tuft structures and brush borders of renal tubules (Fig. 7e). As above, we were able to reproduce in vivo renal regeneration mediated by the replacement of NPCs, and the neoglomerulus acquired a vascular system by becoming integrated into host blood vessels.

**Generation of interspecies chimeric whole nephrons**. By transplanting rat GFP-NPCs into the Six2-iDTR MN of mice, we examined interspecific renal regeneration (Fig. 8f, schematic). The transplanted rat NPCs were subjected to immunostaining 5–7 days after transplantation. As a result, Six2-positive cells of rats were observed in the CM of mice. In these Six2-positive cells, mouse-derived NPCs had been eliminated with DT, thus providing support for the integration of rat-derived NPCs into the mouse-derived CM and ureteric tip (Fig. 8a, left, 8b, and 8c). Pax8 is a transcriptional factor which marked fusion of the renal vesicle with the ureteric tip at the late renal vesicle stage[30]. Further, E-cadherin is expressed in the distal renal epithelium of the fused

nephron with late stage nephron[30] (Fig. 8d). Transplanted rat GFP-NPCs were connected to the mouse ureteric bud (Fig. 8c, d). The anti-vimentin monoclonal antibody V9 was used for species-specific detection in rats but not mice[31]. Immunofluorescence microscopy showed that the transplanted rat GFP-NPCs were stained by rat-specific anti-vimentin antibodies, whereas the mouse ureteric bud was not stained with anti-vimentin antibodies (Fig. 8e). Rat NPCs were replaced by mouse NPCs (Fig. 8b, Supplementary Movie 8). Furthermore, following differentiation into glomeruli and renal tubules, the cells also showed positive staining for WT1 and E-cadherin (Fig. 8c and d). These results demonstrated that renal regeneration mediated by the replacement of NPCs was also possible between different species.

## Discussion

In this study, we used a combination of a specified cell injection method and a drug-induced cell elimination system to exchange native NPCs with exogenous NPCs and generate regenerative nephrons from transplanted NPCs (Fig. 5e and Supplementary Movies 3–6). Most importantly, we succeeded in generating interspecies nephrons in Six2-iDTR mice via replacement of different species of rat NPCs using a drug-induced cell elimination system (Fig. 8). In brief, the present study yielded the following three major findings: (1) nephrons could be generated by transplanting NPCs in the nephrogenic niche of a fetus; (2) elimination of host NPCs in the nephrogenic zone could be achieved by injection with external NPCs, allowing them to develop further to nephrogenesis; and (3) such a system could be applied for generating interspecies nephrons.

We believe that tissue-specific progenitor cells are superior to pluripotent stem cells (PSCs) because, due to their limited differentiation capacity[9], organ regeneration may be restricted, and

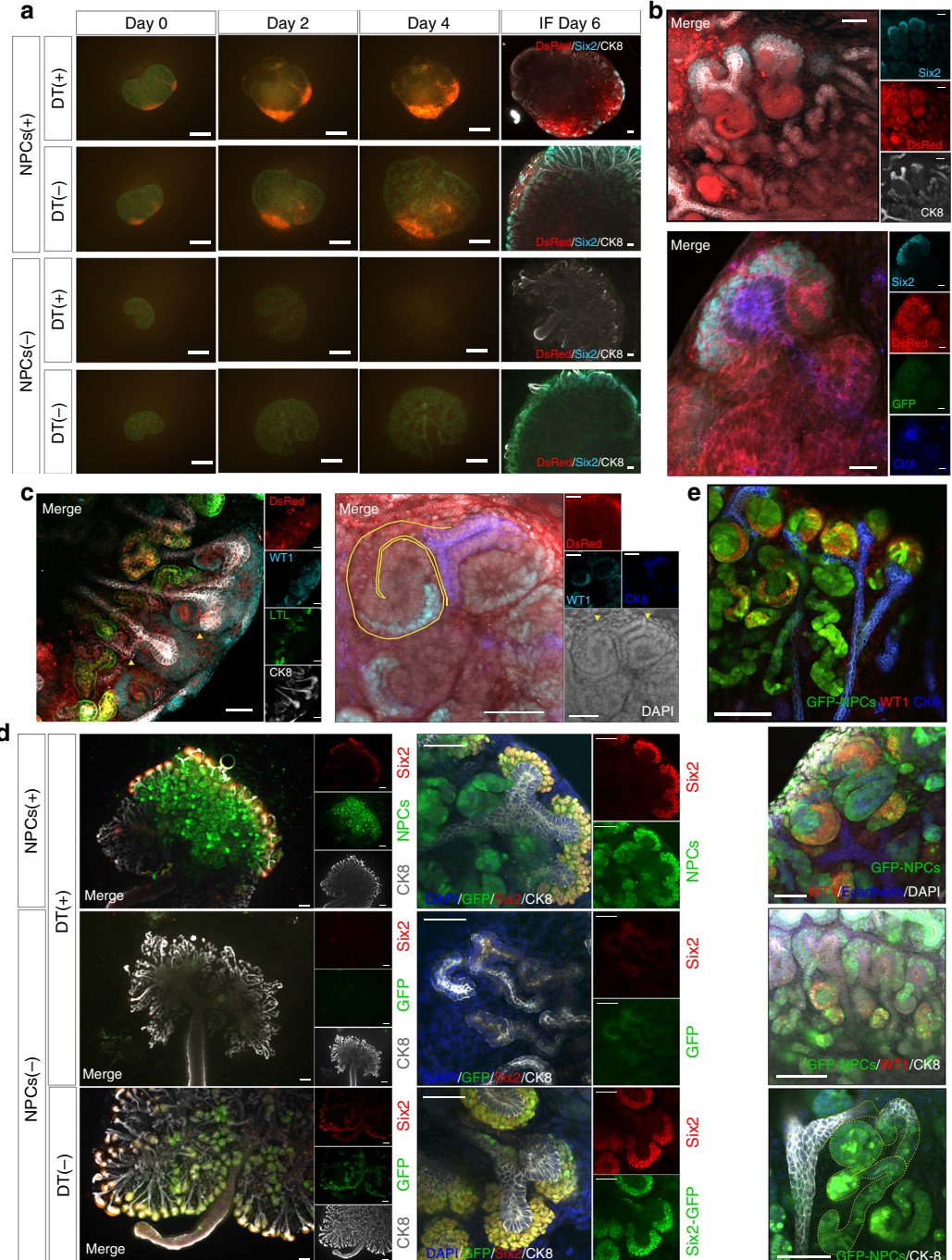

**Fig. 5** Elimination of host NPCs combined with donor NPC transplantation in the nephrogenic zone resulted in regeneration of nephrons. **a** The isolated metanephros (MN) of Six2-iDTR was examined for four groups: (1) transplantation of DsRed-NPCs and daily administration of diphtheria toxin (DT; top column); (2) transplantation of DsRed-NPCs and no administration of DT (second from top column and right); (3) no transplantation of DsRed-NPCs and daily administration of DT (third from top column); (4) no transplantation of DsRed-NPCs and no administration of DT (scale bar, days 0–4: 500 μm; day 6: 100 μm). **b** DsRed-NPCs expressing Six2 were present on the CM (upper column; scale bar, 50 μm). Six2-positive cells expressed DsRed, but not GFP (lower column; scale bar, 20 μm). **c** DsRed-NPCs epithelialized nephrons, including glomeruli (WT1 positive) and renal proximal tubules (LTL positive). Epithelialized nephrons derived from donor cells connected the host collecting ducts (yellow triangle and yellow line; scale bar, 50 μm). **d** Transplanted cells changed DsRed-NPCs to GFP-NPCs. Animals were transplanted with or without GFP-NPCs and treated with or without DT daily (scale bar, 100 μm). Seven days later, samples were immunostained and observed. **d** Transplanted GFP-NPCs were integrated into the CM with Six2 expression and epithelialized vesicles (top). **d** Six2-GFP cells were absent, and poor branching of the ureteric bud, which did elongate but had an extremely deformed shape with many distortions, was observed (second from top). **d** The Six2-iDTR MN harbored GFP-expressing cells, whereas epithelialized nephrons including glomeruli and renal tubules showed no GFP expression (bottom) (left scale bar, 100 μm; right scale bar, 50 μm). **e** Transplanted GFP-NPCs had WT1-positive glomeruli (top column; scale bar, 100 μm) and E-cadherin-positive distal convoluted tubules (second from top; scale bar, 100 μm). These neonephron structures were connected to the ureteric bud sprouting from the host MN (second from bottom, bottom; scale bar, 100 μm)

other organs may not be affected in the chimeras, thereby avoiding ethical concerns. Additionally, several recent protocols have been developed to establish organ-specific progenitor cells from PSCs, such as ES cells and iPS cells, making this system feasible for drug screening and in vivo evaluation in disease models by establishing patient-derived nephrons[15–19].

The advantage of using an elimination system is that donor NPCs can exclusively exploit the developmental program of the developing host embryo, thus making it possible to obtain nephrons consisting purely of the donor cells (Fig. 2b, Fig. 5d, top, Fig. 8a–d, Supplementary Movie 8). Since the drug-induced system can specify the period and parts of elimination, other developmental environments will not be disturbed. In particular, the iDTR model, which represents a DT-mediated cell elimination system, proved suitable for cell replacement because it was very expeditious and efficient in elimination cells[28, 32, 33]. The pancreas, liver, and heart have been reported to be compatible with iDTR-mediated regeneration[34–35-36]. However, application to renal regeneration has not been reported until the present study, in which we attempted to replace progenitor cells as donor cells for the first time. This system takes advantage of the iDTR system to make DT act at specific timing on the target tissue, thereby inducing apoptosis of the target cells in both a time- and tissue-specific manner. We assumed that in order to obtain the normal native organogenesis niche as a biological scaffold, the environment of the niche had to be maintained until just before cell injection. In the present study, we targeted Six2+ NPCs as the cells to be eliminated. In a previous report, adult individuals of the Six2-iDTR strain were used as a pathologic model for chronic kidney disease following acute kidney injury[32]; however, no studies have been performed to analyze the MN during the embryonic stage. Therefore, it is unclear whether Six2-positive (Six2-expressing) cells can be specifically and completely eliminated by the DT-mediated and cell-specific apoptosis induction system. Six2 is a transcription factor expressed within cells present in the CM during nephrogenesis, and Six2-expressing cells are NPCs capable of differentiating into all nephron components, including podocytes, proximal tubules, the loop of Henle, and distal tubules[22]. In contrast, the collecting duct system is derived from a different origin, occurring from the ureteric bud. Six2+ NPCs are multipotent progenitor cells that can differentiate into the nephron unit from the glomerulus to the distal tubules and possess vigorous self-renewing ability[22]. A previous report showed that the Six2-null MN lacked ureteric bud branching and exhibited destruction of the nephrogenic niche[37]. The MN forms through reciprocal interactions between metanephric mesenchyme in the nephrogenic zone and ureteric bud epithelium[38]. Therefore, similar to the Six2-knockout model, elimination of the CM cells at the very beginning was predicted to empty the niche[37].

In Six2-iDTR mice, Six2+ NPCs started disappearing by approximately 36–48 h after DT administration and were eliminated after consecutive DT administration. Intriguingly, even when NPCs present in the CM were eliminated, the ureteric bud elongated to a certain extent, albeit with poor branching and noticeable distortions that would not be seen in normally branching ureteric buds. This may have been attributable to the elimination of the Six2+ NPCs because the branching process is based on the interaction with NPCs[39–40-41]. However, this conditional elimination system did not induce apoptosis of the ureteric bud cells and not disrupt the ureteric tip, which contained the nephrogenesis area. Thus, we believe that the conditional progenitor cell elimination system, which could be switched according to the specific region and timing, played a key role, supporting the importance of replacing cells simultaneously with or immediately after transplantation.

We aimed to confirm whether the NPC eliminating system could be applied to different species to generate human kidneys, in order to overcome the problems with shortage of organs and to establish a noninvasive system for human drug screening and toxicity analysis of human nephrons in nonhuman animals in vivo. Furthermore, this concept will be useful for improving our understanding of human kidney development and disease mechanisms. Thus, we attempted to transplant rat NPCs into the Six2-iDTR MN of mice. Rat NPCs differentiated into nephrons and connected to the mouse ureteric bud and collecting duct (Fig. 8b, d). We succeeded in generating nephrons via an interspecies. These results suggested that this system was an efficiency method to generate chimeras in more disparate species. Hence, this system is expected to be applicable in pigs, whose organ sizes are similar to those of humans[42]. Recently, another team reported that human PSCs injected into pig blastocysts may constitute a first step towards realizing the potential of interspecies regeneration in pigs[43]. Similarly, our strategy using limited chimeras, which targeted the organ niche and lineage progenitors, also suggested the possibility of generating xeno-kidneys in pigs.

For example, NPCs derived from human iPS cells may be transplanted into the nephrogenic zone of the porcine MN, genetically modified to possess a progenitor cell elimination system. Then, modified pig kidneys may function as bioreactors for generation of human nephrons, thus contributing to the creation of chimeric human/pig nephrons. In this study, the function of producing urine was not validated in our rodent system owing to the very small samples. If the system was developed in larger pig fetus kidneys, MNs extracted from the kidney region could be transplanted into adult animals, and the production of urine could be analyzed; such a system may result in improved patient prognosis[7]. However, the iDTR system cannot be directly applied to human cells because human cells generally express DTR (human heparin-binding epidermal growth) factor-like growth factor. Therefore, we have begun to establish other elimination systems using the CreERT2 system, which is a target of Six2+ NPCs[39]. This system uses tamoxifen as an elimination switch and is expected to generally have no effect on human cells.

Subsequently, it will be necessary to determine whether NPCs derived from PSCs arise to differentiate into nephrons in this system. Takasato et al. previously reported that the nephron progenitors derived from human ES cells into re-aggregates of mouse kidney cells were differentiated into nephron and expressed developed kidney markers in vitro[44]. These findings suggested that NPCs derived from human PSCs could be induced by developmental signals in the kidneys of other animals. Another group reported the transplantation of human NPCs derived from iPS cells beneath the kidney capsule of immunodeficient adult mice[45]. As a result, transplanted human NPCs exhibited immature glomerular formation. Thus, if exogenous NPCs are exposed to an appropriate environment (nephrogenesis niche), glomeruli and tubules can self-organize structures derived from NPCs[46]. However, it is still necessary to confirm that neonephrons derived from human NPCs can connect to nonhuman ureteric buds and collecting ducts. In our proof-of-concept study, we found that exogenous NPCs connected with host ureteric buds in conspecies and interspecies analyses; these are basic findings that provide a positive control for evaluating the differentiation of induced NPCs derived from PSCs.

Recently, Yamaguchi et al. reported that mouse pancreata generated in a xenogeneic (rat) environment by interspecies blastocyst complementation resulted in mouse islets cell-engrafted diabetic mice and normalized blood glucose levels[47]. Thus, we will plan to apply this approach to our system. Namely, human kidneys will be generated in a xenogenic (pig)

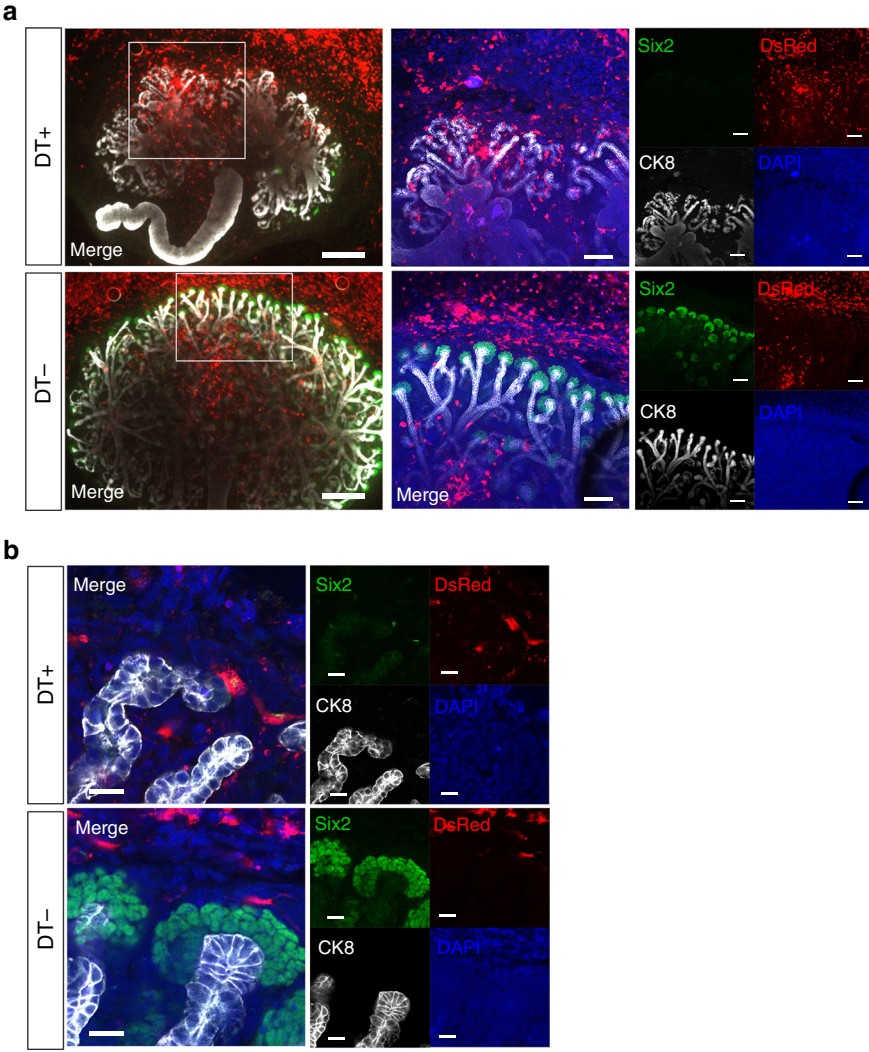

**Fig. 6** Examination of regeneration of nephron progenitor cells using exogenous fibroblasts which is not nephron progenitor cells. **a** DsRed fibroblasts were transplanted into the nephrogenic zone of progenitor elimination model (Six2-iDTR mice, DT+).Neither adherence to the CM nor differentiation into Six2+ cells (upper column) was observed. As a control, DsRed fibroblasts were transplanted into the Six2-iDTR metanephros without DT (lower column). **b** High-magnification images of **a**. **a** left, scale bars: 500 μm; **a** right, scale bars: 100 μm; **b** scale bars: 20 μm; DT+: daily administration of diphtheria toxin, DT−: no administration of diphtheria toxin

environment by elimination of target cells and replacement of exogenous lineage progenitor cells derived from PSCs; the generated human kidneys will be transplanted into patients with end-stage kidney disease. Furthermore, if NPCs derived from PSCs can differentiate into neo-nephrons, this system has the potential for application in drug screening and disease modeling, providing insights into disease mechanisms, drug efficacy, and toxicity in vivo.

In summary, we aimed to utilize the organ niche in fetuses of different species as a platform for organ regeneration; we demonstrated regeneration by transplanting progenitor cells into the organ niche of later stage fetuses. In particular, by conditional elimination of the host progenitor cells in a time- and tissue-specific manner in the nephrogenic niche by the drug-induced cell elimination system in a spatiotemporal manner (summarized in Fig. 8g and Supplementary Movie 9), we successfully replaced the host cells with transplanted cells, thereby regenerating nephrons consisting of donor cells. However, the regenerative kidney remained host tissue, i.e., the ureteric bud lineage and interstitial lineage, and we were unable to replace all kidney structures. The technique developed in the present study may

bridge the gap between the goal to regenerate functional organs in vivo and iPS/ES cell technology in a dish, further improving our understanding of kidney development, disease modeling, and drug screening.

## Methods

**Mouse and rat maintenance and experiments**. Animal experiments followed the Guidelines for the Proper Conduct of Animal Experiments of the Science Council of Japan (2006) and were approved by the Institutional Animal Care and Use Committee of the Jikei University School of Medicine (protocol number: 2016-027, 2016-089, 25-042C4, 27-14, 28-29, II-28-8). All efforts were made to minimize animal suffering. C57BL6/NCrSlc, C57BL/6-Tg(CAG-EGFP) mice (GFP-mice), and Sprague −Dawley-Tg(CAG-EGFP) rats were purchased from SLC Japan (Shizuoka, Japan). C57BL/6-Gt(ROSA)26Sor[tm1(HBEGF)Awai]/J mice (iDTR-mice)[28] and B6.Cg-Tg (CAG-DsRed*MST)1Nagy/J (DsRed-mice) were purchased from Jackson Laboratory (Bar Harbor, ME). Six2-GFP-Cre transgenic mice (Six2-mice) were a gift from McMahon[21]. Six2-mice were crossed with iDTR-mice to obtain bigenic offspring (Six2/DTR-mice). For renal progenitor cell ablation, Six2-DTR mice were subjected to MN isolation, and MNs were administered with variable doses of diphtheria toxin in organ culture. Mice were bred using timed matings; 12 PM on the day of vaginal plug detection was considered 0.5 days postcoitum (dpc). Both male and female mice were used at 10–25 weeks of age.

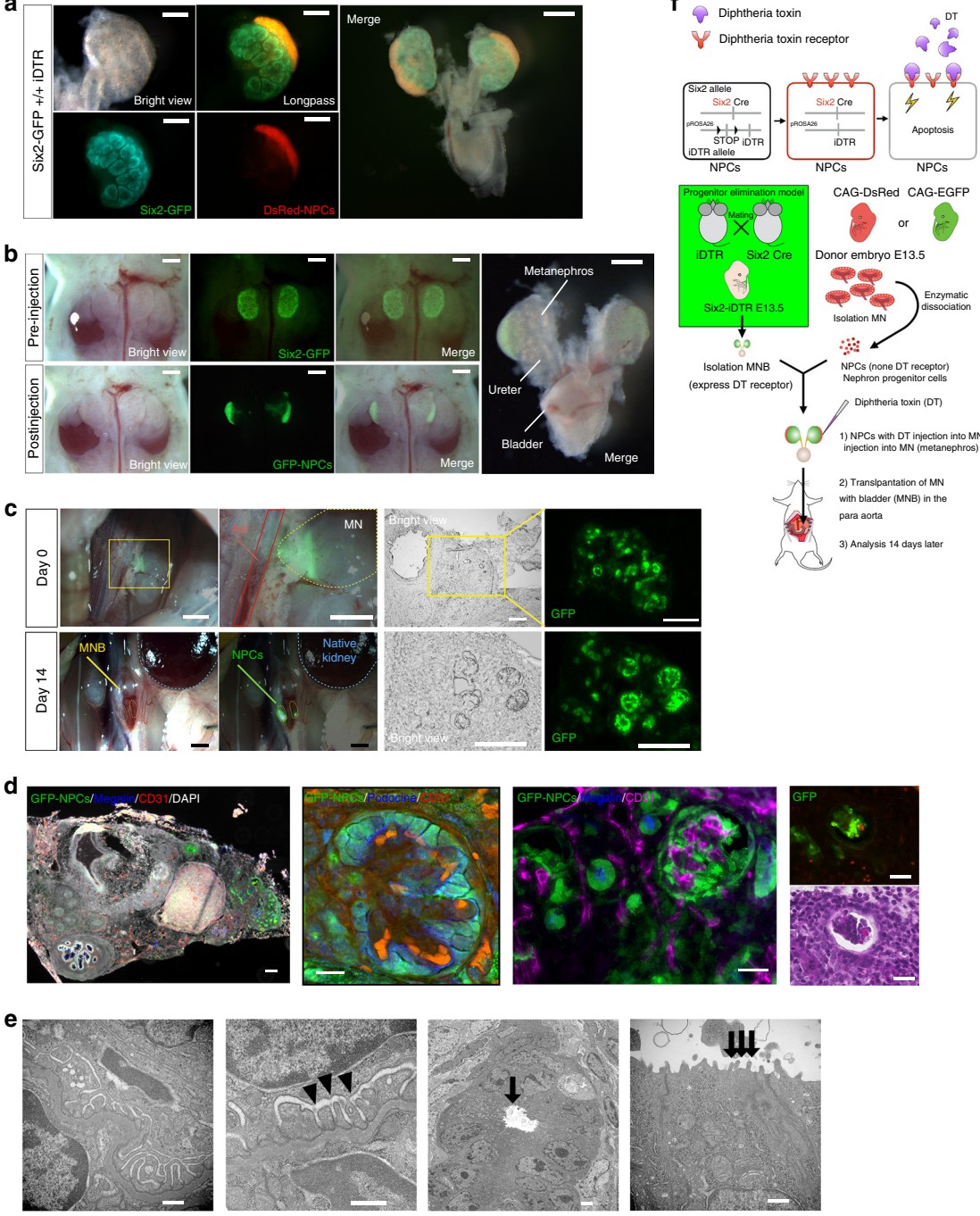

**Fig. 7** Regeneration of the nephron derived from the transplanted cells in vivo and presence of blood vessels in the neoglomerulus. **a** Diphtheria toxin (DT) and nephron progenitor cells (NPCs) were administered under the renal capsule in the cap mesenchyme (CM) of the progenitor ablation model. NPCs showed strong green fluorescent protein (GFP) expression (CAG promoter); therefore, we were unable to observe the NPCs under a fluorescence stereoscopic microscope alongside Six2-GFP (native promoter) after transplanting (scale bar, left: 250 μm, right: 500 μm). **b** The progenitor elimination model into which DT and DsRed-NPCs were transplanted. **c** We transplanted the MN into the vicinity of the aorta in an adult mouse. GFP NPCs and DT were administered under the renal capsule, and GFP expression was evaluated 14 days later (scale bar, left and second from left: 1 mm, right and second from right: 100 μm). **d** From the collected MN, frozen sections were created and analyzed by immunostaining. All of the cells originating in the NPCs were GFP positive. Megalin is a tubular marker, and CD31 is an endothelial marker (scale bar, left column: 100 μm, second from left: 25 μm, second from right: 10 μm). Fourteen days after transplanting, we observed glomeruli originating from transplanted cells within the CM and podocin-positive epithelial cells lining the endothelial cells (second from left). The glomeruli had vascularized in part through blood vessels originating from the host (second from right) and contained red blood cells (right column: 50 μm). **e** Aligned podocytes and glomerular basement membrane in neoglomerulus were observed by transmission electron microscopy (left column, second from left). Slit diaphragm structures were observed (second from left arrowheads). Neo-tubules showed brush borders (second from right, right column arrows). Scale bars: left column, 500 nm; second from left, 500 nm; second from right, 2 μm; right column, 500 nm. **f** Schematic of the experimental protocol

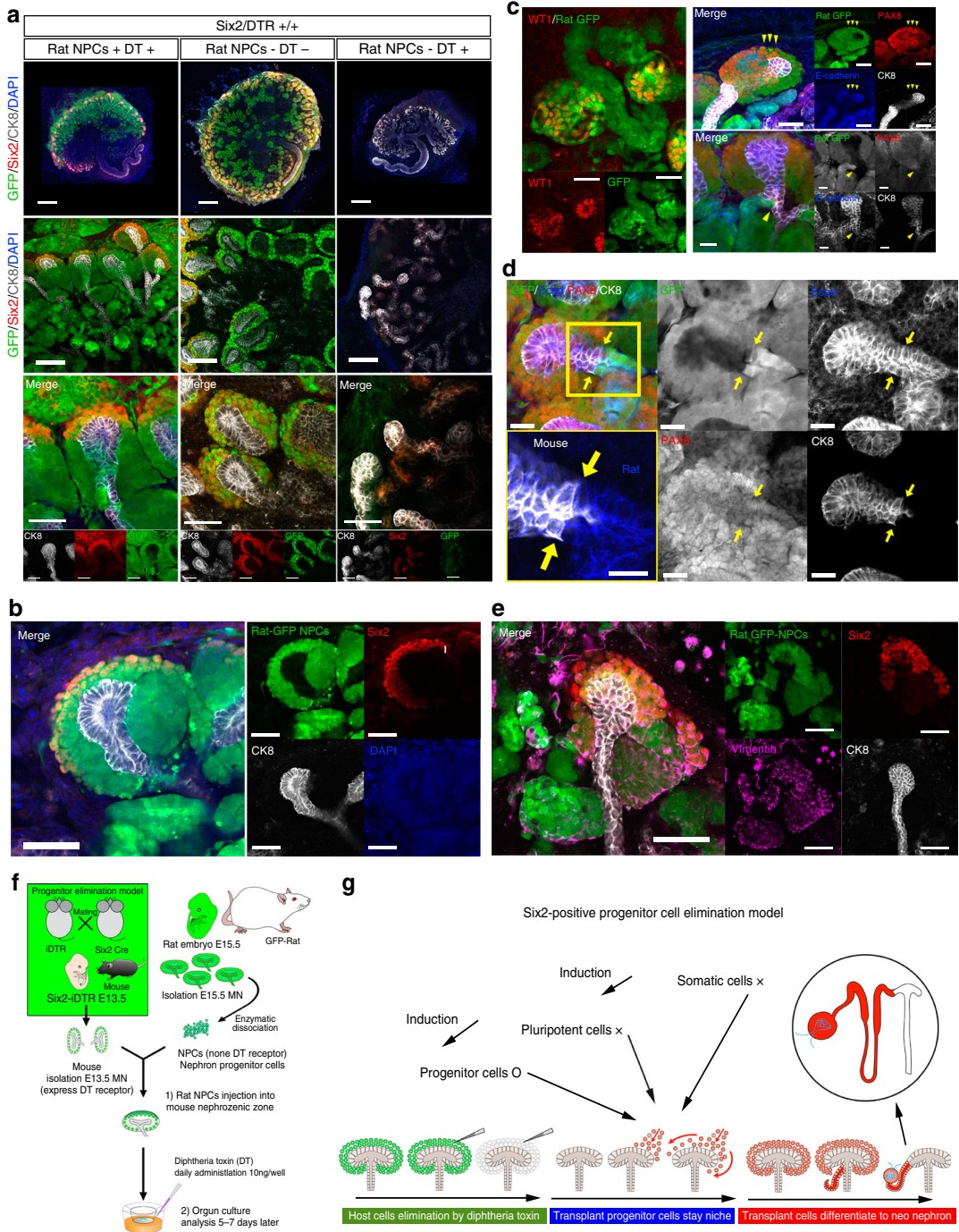

**Fig. 8** Generation of rat nephrons in Six2-iDTR mice via replacement of NPCs using the drug-induced cell elimination system. **a** Transplantation of rat GFP-nephron progenitor cells (NPCs) into the progenitor elimination model (Six2-iDTR) of mouse metanephros (MN). The Six2-iDTR MN were transplanted with or without rat GFP-NPCs and treated with or without diphtheria toxin (DT) daily (scale bars: top column, 500 μm; second from top, 100 μm; bottom, 50 μm). Seven days later, samples were immunostained and observed. Transplanted rat GFP-NPCs with DT were integrated into the mouse cap mesenchyme (CM) and generated GFP-positive epithelialized vesicles (left). The Six2-iDTR MN contained GFP-expressing Six2-positive cells, whereas epithelialized nephrons, including glomeruli and renal tubules, showed no GFP expression (second from left). Six2-GFP cells were absent, and poor branching of the ureteric bud, which did elongate but had an extremely deformed shape with many distortions, was observed (right). **b** Transplanted rat GFP-NPCs with DT significantly replaced host mouse NPCs. GFP-positive pretubular aggregates derived from rat GFP-NPCs were observed (scale bar: 50 μm). **c** Rat GFP-NPCs were differentiated into glomeruli (WT1) and distal renal tubules (E-cadherin, yellow arrowheads) in the Six2-iDTR mouse MN (scale bar: 50 μm). Analysis of Pax8 expression, as a marker of differentiation of the distal nephron segments and metanephric mesenchyme, showed that rat GFP-NPCs were connected continually to the mouse ureteric bud (yellow arrow and dotted line). **d** Rat GFP-NPCs connected mouse ureteric bud was observed (yellow arrows, scale bar: 20 μm). The left lower panels are higher magnification enlargement of the square-enclosed regions within upper panels (scale bar: 20 μm). **e** Rat GFP-NPCs expressed the rat-specific marker vimentin V9, whereas the mouse ureteric bud (CK8) did not express vimentin V9. **f** Schematic of the experimental protocol. **g** Schematic representation of the examination using a drug-induced cell elimination model for NPCs

**Organ culture of isolated MNs**. MNs of mouse embryos from B6 mice or Six2/iDTR mice were used. Isolation of MNs from organ cultures was performed as described by Davies[48]. Briefly, mice were killed by decapitation on E13.5, and MNs were dissected from embryos in Hanks balanced salt solution (Gibco Life Technologies, Grand Island, NY, USA). For in vitro transfilter experiments, isolated MNs of E13.5 embryos were placed at the air–fluid interface on a polycarbonate filter with an average pore size of 0.4 μm (12-mm Transwell; Corning, NY, USA). The medium was α-minimum essential medium (α-MEM; Gibco Life Technologies) supplemented with 20% fetal bovine serum (FBS; Invitrogen) and 1% antibiotic-antimycotic (Gibco Life Technologies). Organs were cultured for 1–7 days at 37 °C with 5% $CO_2$.

**NPCs derived from enzymatic dissociation of MNs**. MNs (E13.5) were prepared from GFP-mice or DsRed-mice. Dissociation of MNs was performed as described by Davies et al.[49]. Briefly, E13.5 mouse MNs were dissected in α-MEM. MNs were then collected into 1 mL prewarmed (37 °C) Accutase (Innovative Cell Technologies) and incubated at 37 °C for 15 min, with manual pipetting every 5 min to aid digestion. MNs were then centrifuged at 300×g for 5 min. Pellets were resuspended in 1 mL phosphate-buffered saline (PBS; Gibco Life Technologies) and manually dissociated to single-cell suspensions. Cells were then passed through a 40-μm cell strainer (BD Falcon) to remove any remaining clumps and centrifuged at 700×g for 3 min. The supernatant was removed completely. Tubes were tapped to mix the pellets and incubated on ice until use.

**DT administration**. To eliminate Six2-positive NPCs in the present study, we used transgenic mice in which cell death may be induced in a temporally and spatially restricted manner. Apoptosis is induced after DT binds to the DTR[28]. Because wild-type mice do not express functional DTRs, we used transgenic mice expressing simian DTR in a Cre-recombinase-inducible manner (iDTR mice)[28]. Six2-Cre mice and iDTR mice were crossbred to generate embryos from which Six2-positive NPCs were eliminated by injection of DT (Wako). DT was dissolved in PBS at 0.1 ng/μL. One hundred microliters of DT (10 ng/well) or vehicle (PBS) was then added to MN culture medium daily for 5–7 days after changing the medium.

**Isolation and engrafting of MNs with bladders (MNBs)**. Pregnant mice (body weight, 20 g) were anesthetized by 1.5% isoflurane (Pfizer) inhalation. Embryos (E13.5) were harvested, and the pregnant mice were then killed immediately by an infusion of pentobarbital (Schering-Plough; 120 mg/kg). All the embryos were killed by decapitation. MNs or MNBs were dissected under a surgical microscope (Leica MZ16FA), as described previously[7]. Isolated MNBs (E13.5) from NPCs injected with DT under renal capsule were transplanted under the retroperitoneal vicinity of the descending aorta by microspatula in anesthetized B6 mice.

**Injection of NPCs into the nephrogenic zone**. B6 (C57BL/6) mice or iDTR mice that were 13.5 days pregnant were killed by dislocation of the cervical vertebrae (Supplemental Movie 2). The fetuses were removed from each specimen through a median incision, along with the uterus, and were then placed in 10-cm dishes containing $Ca^{2+}$- and $Mg^{2+}$-free HBSS (Gibco, Invitrogen; Supplementary Movie 2). The fetuses were then extracted from the isolated uterus and killed by decapitation. The joint parts of the hind legs were slit from the caudal side along the spinal cord. Similarly, the other sides were also slit up to remove the vertebral column from the body. After the pair of MNs became visible in the body, the fetus was fixed with microtweezers. Subsequently, a glass needle filled with cells was inserted into the renal membrane from the renal hilus, and the cells were injected with an injector (SBP-100G-LL; TAKASAGO ELECTRIC, Aichi) or by mouth pipetting (Drummond Scientific Company) under a stereoscopic microscope (Leica MZ16FA). The quantity of one injection was approximately 0.1 μL ($2×10^4$ cells). After the cells were injected carefully to avoid rupturing the renal capsule, the needle was removed when the nephrogenic zone under the renal capsule was filled with a certain number of cells (0.1–0.4 μL, $2–8×10^4$ cells) (Supplementary Movie 2). The transplanted MNs were detached from the fetuses while the ureter and bladder still connected using microtweezers. The MNs were then subjected to organ culture or were transplanted into adult mice. The extracted specimens (MNBs) were in dishes filled with HBSS until use on ice.

**Immunofluorescence**. Immunohistological staining of kidney sections and whole specimens was performed as previously described[48, 49]. Briefly, cryopreserved (7 μm) sections or whole specimens were labeled with primary antibodies, including goat anti-Six2 (Proteintech #11562-1-AP; 1:100 dilution), rabbit anti-WT1 (Santa Cruz Biotechnology #sc-68880; 1:100 dilution), rabbit anti-Podocin (Abcam #ab50339; 1:100 dilution), rat anti-cytokeratin-8 (DSHB #TROMA-1; 1:100 dilution), goat anti-GATA3 (R&D #AF2605; 1:100 dilution), mouse anti-E-cadherin (BD Biosciences #610181; 1:100 dilution), rabbit anti-CD31 (Abcam #ab28364; 1:100 dilution), rabbit anti-Pax8 (Abcam #ab189249; 1:100 dilution), mouse anti-vimentin V9 (Abcam #ab8069; 1:100 dilution), mouse anti-GFP (MBL #M048-3; 1:100 dilution), chicken anti-GFP (Abcam #ab13970; 1:100 dilution), mouse anti-DsRed (Clontech #632393; 1:100 dilution), goat anti-megalin (Santa Cruz Biotechnology #sc-16478; 1:100 dilution), and FITC-conjugated LTL (Vector Laboratories #FL-1321; 1:200 dilution). Slides and whole specimens were

subsequently exposed to corresponding secondary antibodies (Thermo Fisher Scientific; 1:100–200 dilution) and mounted with DAPI- containing Prolong-Gold mounting medium (Thermo Fisher Scientific). Staining was examined by fluorescence microscopy (LSM880 confocal, Zeiss and IX-77, Olympus), and semi-automated quantification and 3D construction were performed using IMARIS software (8 series; Bitplane)[23].

**Measurement of the replacement rate in the cap mesenchyme**. Whole immunostained metanephros were imaged using the Zeiss LSM880 confocal microscope (Zeiss). Confocal immunofluorescence images of GFP (green), Six2 (red), and cytokeratin 8 (white) were analyzed. The images were visualized with Imaris (version 8, Bitplane). The ureteric bud tip was manually captured in the software. The numbers of Six2- and GFP-positive cells were determined by the IMARIS spots function algorithm with the following parameters: XY diameter, 6 μm; background subtraction with automatic threshold applied. Red spots are Six2-positive cells. Yellow spots are double positive cells both Six2 and GFP. The fraction of yellow spots in the CM of the total spots is presented as a percentage.

**Electron microscopy**. For electron microscopic observation, the specimens were fixed with 2.5% glutaraldehyde and 2% paraformaldehyde mixture in 0.1 M phosphate buffer overnight at 4 °C and then postfixed with 1% osmium tetroxide in the same buffer at 4 °C for 2 h. Dehydration was carried out using a graded ethanol series, and the specimens were then placed in propylene oxide and subsequently embedded in Epok 812 (Oken). Ultrathin sections were prepared with a diamond knife, then stained with uranium acetate and lead citrate solution, and observed using an H-7500 electron microscope (Hitachi) at an accelerating voltage of 80 kV.

**Statistical analysis**. Results are expressed as means±standard errors of the means. All assays were repeatable in independent experiments, and the displayed figures are representative of multiple experiments.

**Data availability**. All relevant data are available from the authors on request.

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

## Acknowledgements

We thank E. Kikuchi, H. Gotoh, and M. Ishida for experimental/technical assistance. Six2-Cre mice were gifted from Dr R. Nishinakamura (The Kumamoto University; Institute of Molecular Embryology and Genetics). This work was supported by grants from Japan Agency for Medical Research and Development (AMED) Grant Number 17ek0310006h0002 and Japan Society for the Promotion of Science (JSPS) KAKENHI Grant Number 17K16102, 16H03175. This work also was supported technically by The Jikei University School of Medicine.

## Author contributions

S.Y., S.T., K.M. and T.Y. designed the study. S.Y., S.T., T.F., K.M., S.F. and B.S.K performed the experiments. S.Y., S.F. and S.T. analyzed the data. H.J.O. and T.Y. supervised the study. S.Y. and T.Y. wrote the manuscript.

## Additional information

**Competing interests:** The authors declare no competing financial interests.

