## [Peer Review File · Nature Communications]

Reviewers' comments:

Reviewer #1 (Remarks to the Author):

The authors have established a system by which donor cells can be injected into the nephrogenic zone and native nephron progenitor cells (NPCs) eliminated in time and kidney tubule-specific ways using introduction of the diphtheria toxin receptor into the tubule by using a Six2Cre mouse crossed to an iDTR mouse. When diphtheria toxin is administered to mouse metanephros in vitro there is elimination of endogenous Six2+ NPCs from the nephrogenic niche. This permitted the engraftment of transplanted NPCs from donors. These NPCs developed into mature glomeruli and renal tubules, and blood flow was evaluated following transplantation.

1. Figure 1e: how do the authors know that the GFP in the mosaic pattern in tubule-like structures is not related to cell death of the injected GFP expressing cells, and uptake of that GFP protein into host viable cells.
2. It is stated in lines 182-185 that the regenerative nephron structures that result from injection of donor NPCs into host kidney where tubules have been depleted using diphtheria toxin are "connected to the collecting duct derived from the host ureteric bud". It is not clear from the data presented that the structures are connected to the collecting duct derived from the host ureteric bud. Do the authors mean to say that there is a lumen that connects the nephron and the ureteric bud? This is not demonstrated.
3. It is possible that there is heterogeneity in Six2 expression and some host cells evade the action of diphtheria toxin perhaps because they are beyond the Six2 stage. These cells then go on to maintain partial integrity of the ureteric buds and may contribute to the nephrons that are attributed by the authors to be coming exclusively from the donor cells?
4. As a control, the authors have used fibroblasts transplanted into the Six2 IDTR metanephric mesenchyme and found no Six2+ cells occurring in the conditioned mesenchyme. The results are presented in supplemental figure 3 but they are not well documented. The authors say that they see no Six2+ cells. Are there ureteric structures? Are there any tubular structures over time?
5. The authors conclude that they have produced "the first the first report of renal regeneration focusing on Six2-positive NPCs in which donor NPCs were full replaced". I believe this goes beyond the data. The pictures generally show a combination in a mosaic pattern, and suggest that there might be residual cells that are not killed from the host. Since the diphtheria toxin receptor will only be expressed if Six2 is active, it is possible that some of the structures still include host cells where naturally it would be the case that Six2 expression would be absent.
6. The authors should discuss in more detail how their approach is potentially applicable to the human. The authors do comment to some extent in the discussion but they do not deal with the fact that with the current protocol if extrapolated to human donor cells would result

in a collecting duct system that has derivation from the non-human host.

7. Reference 47 is incomplete.

Reviewer #2 (Remarks to the Author):

In this manuscript, the authors show that the nephron progenitor population of a developing mouse kidney can be replaced by an exogenously supplied nephron progenitor population if ablated. The objective is to illustrate the feasibility of creating a tissue-specific chimeric organ by supplying a replacement progenitor population specific to a particular stage of development in a particular tissue. This overcomes the ethical challenge of supplying cells from another species that may contribute to tissues other than the target tissue. The approach taken here was to isolate cap mesenchyme from transgenically tagged embryonic kidney and introduce this into the kidneys of mice in which the endogenous target population has been deleted in response to diphtheria toxin. The capacity to do this was endowed genetically by generating transgenic mice in which the receptor for the toxin is expressed on the Six2 cap mesenchyme population. In this instance, it is shown that the introduced cap mesenchyme can reconstitute the cap mesenchyme population and go on to generate apparently histologically normal nephrons. While this is not surprising, it has never been demonstrated before. As such, this is a novel and technically challenging advance and the field will be interested to see this. My only reservation lies in the major claim of the manuscript, described as such by the authors, which was that they had shown it to be feasible to combine two species. This last experiment, the rat into mouse chimeric experiment, is the least well characterised. There is no definitive evidence provided that the cells are of rat origin. While the proof that the endogenous mouse cap mesenchyme can be eliminated, the rat cells are not transgenically tagged, so it remains possible that the cells seen are not rat in origin. It would be valuable to find some method to show the difference in species here, but I am not sure how this would be achieved. The authors should be asked to consider what they could use to prove this.

Minor criticisms: While largely well written with the Figures being good illustrations of the claims, there were a few adjectives which appeared somewhat out of place. For example, the term 'injection of a cellular solution' is an unusual way to explain that GFP+ cells were injected.

Response to Reviewer #1

The authors have established a system by which donor cells can be injected into the nephrogenic zone and native nephron progenitor cells (NPCs) eliminated in time and kidney tubule-specific ways using introduction of the diphtheria toxin receptor into the tubule by using a *Six2*Cre mouse crossed to an iDTR mouse. When diphtheria toxin is administered to mouse metanephros in vitro there is elimination of endogenous *Six2*⁺ NPCs from the nephrogenic niche. This permitted the engraftment of transplanted NPCs from donors. These NPCs developed into mature glomeruli and renal tubules, and blood flow was evaluated following transplantation.

1. Figure 1e: how do the authors know that the GFP in the mosaic pattern in tubule-like structures is not related to cell death of the injected GFP expressing cells, and uptake of that GFP protein into host viable cells.

2. It is stated in lines 182-185 that the regenerative nephron structures that result from injection of donor NPCs into host kidney where tubules have been depleted using diphtheria toxin are “connected to the collecting duct derived from the host ureteric bud”. It is not clear from the data presented that the structures are connected to the collecting duct derived from the host ureteric bud. Do the authors mean to say that there is a lumen that connects the nephron and the ureteric bud? This is not demonstrated.

3. It is possible that there is heterogeneity in *Six2* expression and some host cells evade the action of diphtheria toxin perhaps because they are beyond the *Six2* stage. These cells then go on to maintain partial integrity of the ureteric buds and may contribute to the nephrons that are attributed by the authors to be coming exclusively from the donor cells?

4. As a control, the authors have used fibroblasts transplanted into the *Six2* iDTR metanephric mesenchyme and found no *Six2*⁺ cells occurring in the conditioned mesenchyme. The results are presented in supplemental figure 3 but they are not well documented. The authors say that they see no *Six2*⁺ cells. Are there ureteric structures? Are there any tubular structures over time?

5. The authors conclude that they have produced “the first the first report of renal regeneration focusing on Six2-positive NPCs in which donor NPCs were full replaced”. I believe this goes beyond the data. The pictures generally show a combination in a mosaic pattern, and suggest that there might be residual cells that are not killed from the host. Since the diphtheria toxin receptor will only be expressed if Six2 is active, it is possible that some of the structures still include host cells where naturally it would be the case that Six2 expression would be absent.

6. The authors should discuss in more detail how their approach is potentially applicable to the human. The authors do comment to some extent in the discussion but they do not deal with the fact that with the current protocol if extrapolated to human donor cells would result in a collecting duct system that has derivation from the non-human host.

7. Reference 47 is incomplete.

Point-by-point response

Comment 1: Figure 1e: how do the authors know that the GFP in the mosaic pattern in tubule-like structures is not related to cell death of the injected GFP expressing cells, and uptake of that GFP protein into host viable cells.

Response: We thank the reviewer for this comment, and we apologize for the confusion. Because Figure 1-e was low resolution, the mosaic structures were not presented clearly, and the GFP-expressing cells appeared blurred. We performed the experiment again to obtain images of a clear mosaic pattern in the nephron and metanephric mesenchyme. We have provided new, high-resolution images for Figure 1-e accordingly.

Comment 2: It is stated in lines 182-185 that the regenerative nephron structures that result from injection of donor NPCs into host kidney where tubules have been depleted using diphtheria toxin are “connected to the collecting duct derived from the host ureteric bud”. It is not clear from the data presented that the structures are connected to the collecting duct derived from the host ureteric bud. Do the authors mean to say that

there is a lumen that connects the nephron and the ureteric bud? This is not demonstrated.

Response: We thank the reviewer for this important comment. We performed additional experiments to assess the connection between the donor and host. Pax8 is continuously expressed in the connecting segment of the late renal vesicle (Figure 5-c, right). Further, Georgas et al reported that “a continuous basement membrane and a completely patent lumen linking the early nephron with the ureteric epithelium are evident by late comma/early S-shaped body stage, at which time E-cadherin is expressed in the distal renal epithelium of the fused nephron.” (additionally new reference 31, Georgas, et al, Dev Biol, 2009). We also observed E-cadherin is expressed in continuous basement membrane between the nephron and ureteric bud with late stage nephron (Figure 5-d). We believe that transplanted NPCs connected to the collecting duct were derived from the host ureteric bud (Supplementary Movies 5 and 6). Additionally, Figure 5-C right and Figure 5-d showed that the transplanted NPCs connected to the collecting duct were derived from the host ureteric bud.

Comment 3: It is possible that there is heterogeneity in Six2 expression and some host cells evade the action of diphtheria toxin perhaps because they are beyond the Six2 stage. These cells then go on to maintain partial integrity of the ureteric buds and may contribute to the nephrons that are attributed by the authors to be coming exclusively from the donor cells?

Comment 5: The authors conclude that they have produced “the first the first report of renal regeneration focusing on Six2-positive NPCs in which donor NPCs were full replaced”. I believe this goes beyond the data. The pictures generally show a combination in a mosaic pattern, and suggest that there might be residual cells that are not killed from the host. Since the diphtheria toxin receptor will only be expressed if Six2 is active, it is possible that some of the structures still include host cells where naturally it would be the case that Six2 expression would be absent.

Response 3 and 5: We thank the reviewer for this comment. Indeed, there were some Six2-positive cells that could not be eliminated; however, the number of these cells was very small. Nearly all host nephron progenitor cells were eliminated. Figure 3-d (second from top) shows the remaining Six2-positive cells in the two cap mesenchyme area. The few cells that remained after DT administration expressed Six2, but not GFP. We assumed that cells without GFP expression would not show expression of Cre recombinase either and could survive exposure to DT. However, because of the strong toxicity of DT when administered daily, GFP-positive Six2 cells did not survive. The neo-nephron would be expected to show a mosaic pattern if there were many cells remaining. However, we observed pure replacement in the CM after transplantation (Figure 3-d, top right column). The generated pure nephrons, which did not show a mosaic structure, were highly reproducible. We also showed interspecies results in Figure 5b and Supplementary Movie 8. However, we did not show complete elimination using other metanephric mesenchyme markers. Because it is possible that there were surviving cells other than Six2-positive cells, we deleted the words “completely” and “full”.

Comment 4: As a control, the authors have used fibroblasts transplanted into the Six2 IDTR metanephric mesenchyme and found no Six2+ cells occurring in the conditioned mesenchyme. The results are presented in supplemental figure 3 but they are not well documented. The authors say that they see no Six2+ cells. Are there ureteric structures? Are there any tubular structures over time?

Response: We thank the reviewer for this important comment. We performed the experiment again and have modified Supplementary Figure 3 accordingly.

Comment 6: The authors should discuss in more detail how their approach is potentially applicable to the human. The authors do comment to some extent in the discussion but they do not deal with the fact that with the current protocol if extrapolated to human donor cells would result in a collecting duct system that has derivation from the non-human host.

Response: We thank the reviewer for this important comment and agree completely with the reviewer's assessment. We may have written in too much detail of the potential applications in human. In our proof-of-concept study, we found that exogenous NPCs connected with host ureteric buds in conspecifics and interspecies analyses; these are basic findings. We plan to evaluate human-induced NPCs in nonhuman hosts in the future.

However, our findings provide a positive control for evaluating the differentiation of induced NPCs derived from PSCs. This accomplishment may provide insights into methods for isolating nephrons from PSCs. Furthermore, if NPCs derived from PSCs can differentiate into neo-nephrons, this system has the potential to be a platform for drug screening and disease modeling, providing insights into the disease mechanism, drug efficacy, and toxicity in vivo. Human nephrons generated in rodents may be attractive tools for drug screening and disease modeling.

Sharmin et al reported that human NPCs derived from iPSCs changed to immature nephrons below the kidney capsule in immunodeficient adult mice. Thus, exogenous NPCs are exposed to the appropriate environment (nephrogenesis niche), glomeruli and tubules can self-organize structures derived from NPCs. However, in our study, we could not confirm that neo-nephrons derived from human NPCs could connect with nonhuman ureteric buds and collecting ducts. We plan to examine the connections in interspecies studies of human and nonhuman animal in the future. We have added appropriate descriptions to the revised manuscript (blue text, lines 342–354 and 362–364). Additionally, we have added a new reference (Sharmin et al, JASN 2016).

Line 342: “These findings suggested that NPCs derived from human PSCs could be induced by developmental signals in the kidneys of other animals. Another group reported the transplantation of human NPCs derived from iPS cells beneath the kidney capsule of immunodeficient adult mice⁴⁵. As a result, transplanted human NPCs exhibited immature glomerular formation. Thus, if exogenous NPCs are exposed to an appropriate environment (nephrogenesis niche), glomeruli and tubules can self-organize structures derived from NPCs⁴⁶. However, it is still necessary to confirm that neo-nephrons derived from human NPCs can connect to nonhuman ureteric buds and collecting ducts. In further studies, we will transplant mouse ES cell-derived NPCs

(data not shown), induced as previously reported^{16, 18, 19}. In our proof-of-concept study, we found that exogenous NPCs connected with host ureteric buds in conspecies and interspecies analyses; these are basic findings that provide a positive control for evaluating the differentiation of induced NPCs derived from PSCs.”

Line 362: “Furthermore, if NPCs derived from PSCs can differentiate into neo-nephrons, this system has the potential for application in drug screening and disease modeling, providing insights into disease mechanisms, drug efficacy, and toxicity in vivo.”

Comment 7: Reference 47 is incomplete.

Response: We thank the reviewer for this comment. We have rewritten the text for this section and have deleted reference 47 as a result.

Response to Reviewer #2

Comment

In this manuscript, the authors show that the nephron progenitor population of a developing mouse kidney can be replaced by an exogenously supplied nephron progenitor population if ablated. The objective is to illustrate the feasibility of creating a tissue-specific chimeric organ by supplying a replacement progenitor population specific to a particular stage of development in a particular tissue. This overcomes the ethical challenge of supplying cells from another species that may contribute to tissues other than the target tissue. The approach taken here was to isolate cap mesenchyme from transgenically tagged embryonic kidney and introduce this into the kidneys of mice in which the endogenous target population has been deleted in response to diphtheria toxin. The capacity to do this was endowed genetically by generating transgenic mice in which the receptor for the toxin is expressed on the Six2 cap mesenchyme population. In this instance, it is shown that the introduced cap mesenchyme can reconstitute the cap mesenchyme population and go on to generate apparently histologically normal nephrons. While this is not surprising, it has never been demonstrated before. As such, this is a novel and technically challenging advance and the field will be interested to see this. My only reservation lies in the major claim of

the manuscript, described as such by the authors, which was that they had shown it to be feasible to combine two species. This last experience, the rat into mouse chimeric experiment, is the least well characterised. There is no definitive evidence provided that the cells. While the proof that the endogenous mouse cap mesenchyme can be eliminated, the rat cells are not transgenically tagged, so it remains possible that the cells seen are not rat in origin. It would be valuable to find some method to show the difference in species here, but I am not sure how this would be achieved. The authors should be asked to consider what they could use to prove this.

Minor criticisms: While largely well written with the Figures being good illustrations of the claims, there were a few adjectives which appeared somewhat out of place. For example, the term 'injection of a cellular solution' is an unusual way to explain that GFP+ cells were injected.

Response: We thank the reviewer for this comment. We performed additional experiments using CAG-GFP rat cells as a transgenic tag. The Results section and Figure 5 were modified accordingly.

Response of minor criticisms: We thank the reviewer for this important comment. We have revised the text accordingly to avoid awkward language.

REVIEWERS' COMMENTS:

Reviewer #1 (Remarks to the Author):

The authors have adequately addressed my major concerns. There are a few minor things remaining:

1. The authors refer to "DT-ineffective NPCs". This is an awkward way to refer to donor NPCs. Perhaps it should be "DT-unaffected NPCs"
2. "Elimination" misspelled in Fig 4f

REVIEWERS' COMMENTS:

Reviewer #1:

The authors have adequately addressed my major concerns. There are a few minor things remaining:

1. The authors refer to "DT-ineffective NPCs". This is an awkward way to refer to donor NPCs. Perhaps it should be "DT-unaffected NPCs"

2. "Elimination" misspelled in Fig 4f

Point-by-point response

Comment 1: The authors refer to "DT-ineffective NPCs". This is an awkward way to refer to donor NPCs. Perhaps it should be "DT-unaffected NPCs"

Comment 2: "Elimination" misspelled in Fig 4f

Response 1-2: We thank the reviewer for pointing this out. We have revised the text and figure accordingly to correct these errors.

Deletion: Page6 Line175, "DT-ineffective"

Addition: Page6 Line175, "DT-uneffective"

Deletion: Page7 Line269, "DT-ineffective"

Addition: Page7 Line269, "DT-uneffective"

Figure 7f, 8f,

Deletion: Progenitor "Eliminatin" model

Addition: Progenitor "Elimination" model